# Torque Teno Sus Virus 1: A Potential Surrogate Pathogen to Study Pig-Transmitted Transboundary Animal Diseases

**DOI:** 10.3390/v16091397

**Published:** 2024-08-31

**Authors:** Xiaolong Li, Brandon M. Parker, Raoul K. Boughton, James C. Beasley, Timothy J. Smyser, James D. Austin, Kim M. Pepin, Ryan S. Miller, Kurt C. Vercauteren, Samantha M. Wisely

**Affiliations:** 1Department of Wildlife Ecology and Conservation, University of Florida, Gainesville, FL 32611, USA; xiaolong.li@ufl.edu (X.L.); parker.brandon@epa.gov (B.M.P.);; 2Buck Island Ranch, Archbold Biological Station, Lake Placid, FL 33960, USA; raoul.boughton@mosaicco.com; 3Savannah River Ecology Laboratory, Warnell School of Forestry and Natural Resources, University of Georgia, Athens, GA 30602, USA; beasley@srel.uga.edu; 4National Wildlife Research Center, United States Department of Agriculture, Animal and Plant Health Inspection Service, Wildlife Services, Fort Collins, CO 80526, USAkim.m.pepin@usda.gov (K.M.P.);; 5Center for Epidemiology and Animal Health, United States Department of Agriculture, Animal and Plant Health Inspection Service, Veterinary Services, Fort Collins, CO 80525, USA

**Keywords:** Torque teno sus virus, surrogate pathogen, molecular epidemiological analysis, pig-transmitted diseases

## Abstract

Understanding the epidemiology and transmission dynamics of transboundary animal diseases (TADs) among wild pigs (*Sus scrofa*) will aid in preventing the introduction or containment of TADs among wild populations. Given the challenges associated with studying TADs in free-ranging populations, a surrogate pathogen system may predict how pathogens may circulate and be maintained within wild free-ranging swine populations, how they may spill over into domestic populations, and how management actions may impact transmission. We assessed the suitability of Torque teno sus virus 1 (TTSuV1) to serve as a surrogate pathogen for molecular epidemiological studies in wild pigs by investigating the prevalence, persistence, correlation with host health status and genetic variability at two study areas: Archbold’s Buck Island Ranch in Florida and Savannah River Site in South Carolina. We then conducted a molecular epidemiological case study within Archbold’s Buck Island Ranch site to determine how analysis of this pathogen could inform transmission dynamics of a directly transmitted virus. Prevalence was high in both study areas (40%, *n* = 190), and phylogenetic analyses revealed high levels of genetic variability within and between study areas. Our case study showed that pairwise host relatedness and geographic distance were highly correlated to pairwise viral genetic similarity. Molecular epidemiological analyses revealed a distinct pattern of direct transmission from pig to pig occurring within and between family groups. Our results suggest that TTSuV1 is highly suitable for molecular epidemiological analyses and will be useful for future studies of transmission dynamics in wild free-ranging pigs.

## 1. Introduction

Transboundary animal diseases (TADs) of domestic pigs (*Sus scrofa domesticus*) have increased in notoriety because pork is the most consumed meat by humans [1], and multiple pig-specific TADs have rapidly emerged and spread widely, including classical swine fever (CSF), African swine fever (ASF), porcine epidemic diarrhea (PED), porcine reproductive and respiratory syndrome (PRRS), and foot and mouth disease (FMD) [2]. The spread of TADs among domestic pigs has had significant economic impacts. For example, outbreaks of ASF in China and neighboring countries have led to US$130 billion in losses, resulting in the restructuring of global export markets for pork products [3]. Similarly, an outbreak of CSF in the Netherlands in 1997 led to the destruction of 10 million pigs, with economic damages of US$2.3 billion [4]. In 2001, an outbreak of foot and mouth disease in the United Kingdom led to costs of US$4.4 billion [5].

Adding to the risk of pig-borne TADs, wild pigs (*S. scrofa*), which have a global distribution, contribute to the maintenance and spread of pig TADs [6] because they are reservoirs for multiple pathogens of importance to the global pork industry [7]. In Europe, for example, the establishment of ASF among wild boar populations is maintaining the presence of the disease within the landscape, thus posing a persistent spillover risk to domestic herds [8]. Similarly, in the US, invasive wild pigs have been found to carry various pathogens, such as pseudorabies virus, *Brucella* spp., *Mycobacterium bovis*, and vesicular stomatitis virus [9,10]. Long-term eradication campaigns in the US have eliminated pseudorabies and brucellosis from the domestic swine industry; however, both diseases are prevalent among wild pigs, and spillover to domestic herds periodically occurs and poses a continual risk to the pork industry [11].

Understanding the potential transmission dynamics of TADs among wild pigs is essential for controlling the introduction of TADs into wild pig populations [12]. However, given the challenges of conducting studies of TADs in wild free-ranging populations during outbreaks, a surrogate pathogen system for pathogen dynamics studies would provide a model system to better predict how these pathogens may circulate and be maintained among wild populations, how TADs may spillover into commercial operations [8], and how disease spread may be mitigated.

The ideal surrogate pathogen would have several properties to serve as a workable model system. To effectively and safely study the virus in laboratory settings, it is important that the associated clinical illness be minimal. A virus inducing minimal health impacts on hosts would allow studies on wild free-ranging populations to be tractable. In addition, the pathogen should have an apparent prevalence that allows for consistent observation and provides sufficient statistical power for the estimation of common epidemiological parameters [13]. Also, the pathogen’s ability to infect the cells of a host continuously without being cleared is helpful for identifying transmission routes and increasing detection probability; transient infections are challenging to detect and observe. An ideal surrogate should also have known modes of transmission to accurately represent how TADs with similar transmission modes may spread. Finally, the model virus should have high genetic variability to allow transmission networks to be created based on pathogen phylogenetics [14,15]. These transmission networks can then empirically test models based on proximity [16].

Torque teno viruses (TTVs) are a group of non-enveloped, single-stranded DNA viruses that belong to the family *Anelloviridae* [17]. Originally, these viruses were discovered in a human patient in Japan but have since been found in domestic and wild animals, including dogs (*Canis lupus familiaris*), cats (*Felis catus*), livestock (pigs, sheep [*Ovis aries*], cattle [*Bos taurus*], camels [*Camelus*], and poultry [*Gallus gallus domesticus*]), American badgers (*Taxidea taxus*), tree shrews (*Tupaia belangeri*), bats (*Chiroptera*), and wild pigs [18,19,20,21,22]. These newly discovered members of the *Anelloviridae* family were grouped into 30 genera based primarily on host species [23]. Swine TTVs are classified into 2 genera: Iotatorqueviruses (including species of Torque teno sus virus 1 [TTSuV1]) and Kappatorqueviruses (including species of Torque teno sus virus 2 [TTSuV2]) [24]. Both genera are prevalent in domestic (46–94%) [25,26,27] and wild pigs (32–66%) [28,29].

A previous study on the genetic characteristics of TTSuVs showed that the complete genomes of TTSuV1 had more variable positions than that of TTSuV2 (46.3% versus 23.9%) [30], with the former also possessing higher genetic variability (>30% versus <15%, respectively). Currently, four subtypes of TTSuV1 are recognized (TTSuV1a, b, c, and d) [30,31,32]. Given the high genetic variability of TTSuV1, further classification of subtypes has been proposed [24]. Co-infections of multiple species/subtypes of TTSuV1s within a single pig are commonly observed in domestic herds [31,33]. The high genetic variability allows for various analysis approaches, such as transmission networks and molecular epidemiological models, to be applied at the individual scale or as a comparison of regional to global phylodynamics across geographic scales.

Considering the reported high prevalence and genetic variability of TTSuV1, as well as its non-pathogenicity in all animal species observed to date [24], we assessed its potential utility as a surrogate pathogen to study wild pig TAD transmission in free-ranging populations. To accomplish this aim, we evaluated the prevalence and the phylogenetic suitability of TTSuV1 for molecular epidemiological studies and indirectly inferred the persistence of infection, host health, and transmission factors in two populations of free-ranging wild pigs. In addition, we conducted a case study within one of the wild pig populations to investigate the potential insights that molecular epidemiological analysis of TTSuV1 could provide for transmission studies. Our hypotheses posited that closely related pigs, especially females, and juveniles, would exhibit spatial proximity given their matriarchal social structure [34] and that this spatial proximity would lead to transmission within family groups.

## 2. Materials and Methods

### 2.1. Field and Laboratory Protocols

We captured wild pigs using baited traps at 2 sites in the USA from February 2017 to February 2018 (Figure 1). Capture locations were non-random and located based on environmental signs of wild pig activity or documentation of local presence using game cameras. Wild pigs captured at Archbold’s Buck Island Ranch (ABIR; Lake Placid, FL, USA) were chemically immobilized following procedures from Kreeger and Arnemo (2012) (University of Florida Institutional Animal Care and Use Committee [IACUC] Protocol 201808495), whereas wild pigs captured at Savannah River Site (SRS; Aiken, SC, USA) were either lethally sampled (University of Georgia [UGA] IACUC Protocol A2015 12-017) or chemically immobilized at point of capture (UGA IACUC Protocol A2015 05-004). We collected whole blood (stored in ethylenediaminetetraacetic acid [EDTA] or mammalian lysis buffer) and oral, nasal, and genital swabs (stored in mammalian lysis buffer) from all wild pigs. Sex and age class, estimated based on size at ABIR and by tooth eruption [35] at SRS, were recorded from both sites, and location and morphological data (body mass and body measurements) were captured. We tagged and subsequently released animals at the point of capture. Biological samples were stored at −20 °C for 24–48 months until extracted.

Viral DNA was extracted from whole blood using the Gentra Puregene Blood Kit (Qiagen, Germantown, TN, USA) according to the manufacturer’s ‘compromised blood’ protocol. Whole blood in mammalian lysis buffer was used for screening animals only when blood in EDTA was not collected for a given animal. Swab samples were extracted using the DNeasy Blood and Tissue Kit (Qiagen, Germantown, TN, USA) following the manufacturer’s protocol for the purification of blood or animal cells. Extracted DNA was quantified using a Nanodrop One spectrophotometer (Thermo Fisher Scientific, Waltham, MA, USA). Any samples with a concentration >100 ng/μL were diluted to 75 ng/μL to prevent PCR inhibition. A one-step PCR assay was conducted using primers previously described (forward: 5′CGGGTTCAGGAGGCTCAAT3′ and reverse: 5′TCTACGTCCCCTCTACGG3′) [27] to detect TTSuV1 based on a 678 base pair (bp) sequence that encompasses portions of the untranslated region (UTR) and open reading frame (ORF) 1 and the entire ORF2. PCR products were visualized on 2% agarose gels to identify TTSuV1-positive samples.

### 2.2. Sequence Analyses

All PCR products of the appropriate size were submitted to Eurofins Genomics LLC (Louisville, KY, USA) for bidirectional Sanger sequencing. We manually edited and assembled sequences in Geneious Prime v2023.1.2 (Biomatters, Auckland, New Zealand) and performed Basic Local Alignment Search Tool (BLAST) from the National Center for Biotechnology Information (NCBI; Bethesda, Rockville, MD, USA) to confirm that assemblies were TTSuV1. Viral sequences from each study site were aligned using MUSCLE embedded in Geneious Prime separately, and a pairwise percentage identity (PI) matrix was constructed to measure the percentage of base pairs shared between sequences. We used a classification system proposed by Webb et al. [24], and assemblies with a PI > 85% compared to the BLAST hits were considered TTSuV1. Within confirmed TTSuV1 assemblies, pairwise identities between 85 and 95% were used as benchmark values for the identification of subtypes and identities > 95% for variants, with the caveat that this classification was based on whole genome sequences. Duplicated viral sequences (from serial infection and the same infection found in multiple tissues) were excluded from alignment and subsequent phylogenetic analysis.

### 2.3. Population Analyses

#### 2.3.1. Prevalence and Hot Spots of TTSuV1 Infections among Wild Pigs

An individual was considered TTSuV1-positive if the BLAST-confirmed sequence was found in at least one biological sample collected from a given wild pig. Apparent prevalence was determined as the percentage of individuals that were TTSuV1 positive in proportion to the number of individuals tested. The prevalence of TTSuV1 was calculated separately for each sample type and cumulatively for the host population at each site. The difference in apparent prevalence between sites and sample types was evaluated via a set of chi-square tests or Fisher’s exact tests for scenarios where one or more values in a contingency table were less than 5. Wilson confidence limits were constructed for apparent prevalence, as described in [36]. We also analyzed the apparent prevalence among age classes and between sexes for each site separately (due to differences in age estimation and classification during collection between study sites). The resulting *p*-values were adjusted with Bonferroni correction to control the experiment-wide error rate.

To identify hotspots of TTSuV1-positive wild pigs at the two study sites, we performed a kernel density estimation (KDE) analysis using ArcGIS 10.8.2 (ESRI; Redlands, CA, USA). The densities of observed infected pigs (*n* = 74) were estimated by KDE, and smoothed raster surfaces with density values were generated as output. To set up the KDE analysis, we applied the optimal bandwidth function (*h_opt_*) to calculate the bandwidth used in the kernel function as follows [37]:hopt=23n14σ
where n denotes the sample size (i.e., the number of capture locations), and σ denotes the standard distance among capture locations. Standard distance is a measure of the dispersion of locations around their geographical mean center (or spatial mean) and was calculated with the spatial statistics toolbox in ArcGIS 10.8.2. Due to the relatively small geography of the study sites, the grid cell was set as 100 × 100 m. Following Nelson and Boots [38], we reclassified the estimated density values and used the upper 25%, 10%, and 5% values to define and visualize the hotspots of infected wild pigs.

#### 2.3.2. Persistence of TTSuV1 Infection

Individual pigs were considered to have a persistent infection (>30 days after the prior encounter) if they tested positive for the same variant of TTSuV1 at subsequent recaptures. If viral sequences from a recaptured host had ≥95% PI, then they were considered the same variant and, therefore, a persistent infection; otherwise, the viral sequences were considered different variants and assumed to be independently acquired [24].

#### 2.3.3. TTSuV1 Infection and Body Condition of Wild Pigs

To determine if body condition was correlated with the infection status of TTSuv1 in wild pigs, we compared the body condition of infected and uninfected pigs. Compromised body condition, characterized by factors such as weight loss and decreased fat reserves, is often indicative of compromised immune-system responses in wildlife populations [39,40]. To estimate body condition, mass and morphological data (body measurements detailed in Table 1) were used to calculate the scaled mass index (SMI) of wild pigs from both sites separately. Wild pig SMI was calculated as follows:$$SMI=M_{i}{\left[\frac{L_{o}}{L_{i}}\right]}^{b_{SMA}}$$
where *M_i_* and *L_i_* are the body mass and body measurement of individual *i* respectively; *L_o_* is the average body measure of the population; and *b_SMA_* is the slope of the ordinary least squares regression divided by the Pearson’s correlation coefficient (*r*) of the measurement that best correlates with weight [41]. An ANOVA was used to test for the effect of age class, sex, and TTSuV1 infection status on wild pigs.

#### 2.3.4. Genetic Divergence of TTSuV1 over Space

We used Mantel correlograms, which are graphical representations of changing Mantel correlations over different distance intervals, to evaluate the extent of spatial structure on the genetic divergence of TTSuV1 at the two sites. A Mantel test is a common approach in ecology and biology to estimate the correlation between two distance matrices, e.g., geographic distance and genetic distance [42]. By dividing the geographic distance into multiple distance classes and performing a Mantel test at each distance class, the spatial extent of the patterns of correlation can be characterized in a Mantel correlogram. In this study, we used two sets of breakpoints to define distance classes according to the different landscape scales at the two study sites. The breakpoints for ABIR included 500, 1000, 1500, 2000, 4000, and 8000 m. Those for SRS were 500, 1500, 3000, 6000, 9000, 12,000, 15,000, and 20,000 m.

Sequences obtained from TTSuV1-positive pigs were used to generate pair-wise genetic distance matrices with the F84 model [43] for ABIR and SRS, respectively. A geographic distance matrix between traps was calculated by employing the haversine formula. All these steps were performed in R (version 4.1.1) [44]; the ‘ape’ package [45] was used for calculating genetic distance, and the ‘vegan’ package [46] was used for constructing Mantel correlograms.

### 2.4. Phylogenetic Analysis

Substitution saturation refers to instances where multiple substitutions have occurred at the same nucleotide positions in an alignment, resulting in a loss of phylogenetic information [47,48]. To ensure that our targeted sequence had adequate phylogenetic information, we tested this DNA segment for substitution saturation via Xia’s test [48] using the DAMBE program [49]. We compared pairwise sequences by calculating the genetic distance and the proportion of observed transitions and transversions in each pair of sequences. Then, two curves representing the observed proportions of transitions or transversions, respectively, were plotted as a function of pairwise genetic distance. If both curves increase with genetic distance, it suggests that the number of transitions and transversions is proportional to the genetic distance, and the sequences are not saturated. Substitution saturation occurs if one or both curves reach a plateau, suggesting that no new information is provided by additional substitutions.

A phylogenetic tree was constructed using TTSuV1 sequences isolated from both sites as well as reference sequences reported in the literature [27,30,31] to assign each sequence a subtype of TTSuV1. We used a neighbor-joining algorithm to construct the tree and estimated the confidence of internal branches with 1000 bootstrap replicates using MEGA version 11 [50].

### 2.5. Host Relatedness and Viral Transmission: A Case Study at ABIR

If TTSuV1 was suitable for molecular epidemiological analyses, we hypothesized that our analyses should discern direct transmission of the virus from pig to pig. Because TTSuV1 is directly transmitted, we further hypothesized that the virus would be more likely to be transmitted among closely related pigs within a family group. To test this hypothesis, we estimated the genetic relatedness among wild pigs and the virus similarity among the same pig hosts. We then conducted a molecular epidemiological case study, where we identified pairs of pigs inferred to have had direct transmission of the virus. We characterized the genetic and geographic relationships of these pairs to determine if direct transmission occurred within family groups more frequently than among unrelated individuals.

#### 2.5.1. Host Genotyping Data

Genomic DNA was extracted from tissue samples collected from the pinna of wild pigs using a magnetic bead extraction protocol (MagMax DNA, Thermo Fisher Scientific, Waltham, MA, USA). Wild pigs were then genotyped with GeneSeek’s Genomic Profiler for Porcine HD (an Illumina [San Diego, CA, USA] BeadChip Array, licensed exclusively to GeneSeek, a Neogen Corporation, [Lansing, MI, USA] [51], yielding 62,128 biallelic single nucleotide polymorphic (SNP) loci mapped to autosomes as described on the Sscrofa 11.1 genome assembly [52]. Genotypes were screened using standard quality control metrics, pruning loci with call rates <95%, loci with minor allele frequencies <5%, and individual genotypes with call rates <95%. With high-density SNP arrays, there is an expectation of linkage among proximate loci within chromosomes. Accordingly, loci were evaluated for linkage disequilibrium (LD), pruning a single locus from closely linked SNP dyads (i.e., *r*^2^ > 0.5) by screening loci within a genomic window size of 50 loci and a sliding window increment of 5 loci. After implementing these quality control and LD pruning measures, 96 wild pigs genotyped at 29,863 loci were available for analysis.

#### 2.5.2. Construction of Host Relatedness Matrix and Genetic Distance Matrix

As a means of characterizing genetic differences among wild pigs, pairwise estimates of relatedness (*r*) were calculated using methods implemented with R packages ‘PC-AiR’ [53] and ‘PC-Relate’ [54] that correct for the underlying genetic structure to produce unbiased estimates of relatedness. Specifically, this approach first uses a nested Principal Component Analysis (PCA) to characterize latent population genetic structure and, second, estimates relatedness while correcting for the underlying structure. Evaluation of eigenvalue scree plots from the nested PCA and diagonal elements in the associated relatedness matrix (relatedness of an individual to itself when correcting for population structure) [55] demonstrated that 2 principal components (PCs) were sufficient to explain the underlying genetic structure and, thus, were carried forward to estimate relatedness.

As a complementary approach for the description of pairwise genetic differences among wild pigs, an independent PCA was conducted with R package ‘adegenet’ [56], similarly using eigenvalue scree plots to determine the number of PCs necessary to describe the genetic structure within the sample. Genetic distances among all combinations of wild pigs were then estimated as the Euclidean distance across the 4 retained PC axes, as informed by the scree plot [57], and calculated with R package ‘geosphere version 1.5-18′ [58].

#### 2.5.3. Correlation Analysis of Genetic Distance Matrices of Hosts and Viruses

The genetic divergence of wild pigs sampled at ABIR over space was first tested and visualized using the Mantel correlogram. Then, a subset of the host genetic distance matrices for infected wild pigs was obtained and used for correlation analysis against the viral genetic distance matrix via the Mantel correlogram.

#### 2.5.4. Inference of Host-To-Host Viral Transmission

As a first step to infer transmission events based on viral sequences, we reconstructed a time-dated phylogenetic tree using the BEAST 2 software package [59]. Hasegawa-Kishino-Yano (HKY) and coalescent constant population were selected as the substitution and population models, respectively. We ran three molecular clock models separately (i.e., strict clock, optimized relaxed clock, and random local clock) and chose the best-fit model by estimating the marginal likelihood through nested sampling and calculating the Bayes factor [60,61]. The prior for the mean clock rate was set as 5.0 × 10^−4^ substitutions site^−1^ year^−1^ based on a previous study [28]. We applied a Markov chain Monte Carlo (MCMC) chain length of 80,000,000 to ensure sufficient mixing and checked the convergence of all parameters in Tracer version 1.7.2 with the threshold of effective sample size (ESS) greater than 200. The maximum clade credibility (MCC) tree was constructed in TreeAnnotator version 2.7.5 with a 10% burn-in.

With the dated phylogeny reconstructed above as an input, we then inferred the transmission tree using TransPhylo (v1.4.5), a Bayesian approach to infer transmission events based on a combined model of between-host transmission and within-host evolution of pathogens [62]. As the generation time (i.e., the average time interval between the infection of one individual and the subsequent infections transmitted from such individual) of TTSuV1 was not reported in the literature, we used a similar circular DNA virus in pigs, porcine circovirus type2 (PCV2), as a proxy and generated a gamma distribution with the mean of 18 days to represent the generation time [63]. Along with the inference of the transmission tree, several key parameters were estimated using 1,000,000 iterations. These parameters include the sampling proportion (pi), the within-host coalescent time (Neg), and two parameters related to the offspring distribution describing the number of secondary cases resulting from each infection. We removed the initial 10% of the MCMC chains as burn-in, and all MCMC chains were run until convergence was reached as measured by EES (>200) and by visually inspecting the MCMC traces.

Pairs of infector-infectee host individuals from our data set inferred by the transmission tree were recorded, and the corresponding sex, age, capture location, capture date, and the genetic relatedness of pigs were retrieved from the metadata for comparison.

## 3. Results

### 3.1. Prevalence and Hot Spots of TTSuV1 Infections among Wild Pigs

A total of 171 unique wild pigs were captured (100 and 71 from ABIR and SRS, respectively). Of these wild pigs, 15 were captured twice and 2 three times, which resulted in a total of 184 whole blood samples, 190 oral samples, 190 nasal samples, and 188 genital samples (Table 2). The apparent prevalence of TTSuV1 infections among wild pigs was 39.2% (40/102) and 38.6% (34/88) at ABIR and SRS, respectively, with no significant difference between sites (*p* > 0.05). TTSuV1 DNA was found in all sample types tested but was most frequently found in whole blood samples. Mucosal samples from ABIR yielded no positive results, yet TTSuV1 was detected in mucosal samples from SRS. We found that nasal and oral swabs from SRS had significantly higher prevalence (*p* < 0.05) than that from ABIR. In comparison, genital swabs had a marginally higher prevalence (*p* = 0.051) at SRS than at ABIR, yet we found no difference in the prevalence of TTSuV1 in whole blood between study areas. Whole blood samples had the highest prevalence across all sample types at both sites, and 97.3% (72/74) of TTSuV1-positive animals were diagnosed through screening whole blood alone. For the two positive animals that were negative based on blood tests, TTSuV1 was detected either in oral or nasal samples. Mucosal tissues had significantly lower prevalence (*p* < 0.001) than blood across both sites.

We detected a difference in TTSuV1 prevalence among age classes at both sites. Of the 63 adult wild pigs captured at ABIR, 65.1% (41/63) were positive for TTSuV1—significantly higher (*p* < 0.001) than juveniles (17.6% [6/34]) at the same site. The difference in prevalence between these two age groups was not significant at SRS (37.9% [adults] vs. 68.8% [juveniles]; *p* > 0.05). We found no difference in prevalence between males and females at both sites (*p* > 0.05).

Kernel density estimation of the infected wild pig cases identified hotspot areas in the northeastern corner of ABIR (Figure 2A). Since the SRS study area was substantially larger than ABIR, and most capture locations were clustered in the central portion of the site, the hotspots identified by KDE were located in the center of the capture locations (Figure 2B).

### 3.2. Persistence of TTSuV1 Infection

Across both sites, 14 wild pigs were recaptured, of which 9 were infected during only one capture and 3 were infected at both captures. The overall interval between recaptures ranged from 48 to 356 days, with an average of 182 days. The time intervals for the 3 animals that were positive upon both captures were 48, 197, and 248 days. One animal had a persistent TTSuV1 infection with the same variant (1.2 km apart between captures). The other two animals that tested positive for TTSuV1 during multiple captures were infected with different subtypes of TTSuV1 between captures (84% and 82% genetic identity), suggesting reinfection rather than persistence.

One wild pig was initially infected with TTSuV1, but no TTSuV1 was detected at recapture (the interval between the two captures was 220 days), suggesting the animal cleared the infection. Conversely, eight wild pigs that initially tested negative were positive for TTSuV1 when resampled (intervals range = 67–259 days).

### 3.3. Co-Infections

Co-infections with multiple TTSuV1 variants/subtypes were detected in 12.7% (9/71) of infected wild pigs captured at SRS. No co-infection was found in wild pig samples from ABIR. Of the 9 animals with co-infections, 8 were suspected to have multiple TTSuV1 variants (percentage identity <95%) present in different tissues, and one was infected with different subtypes of TTSuV1 simultaneously. For 8 of 9 animals, one infection was detected in whole blood and the other in mucosal tissue. Only one wild pig was positive for the same variant of TTSuV1 detected in different mucosal tissues, not in whole blood.

### 3.4. TTSuV1 Infection and Body Condition of Wild Pigs

Through testing the correlation between body mass and multiple body measurements for each site, we found chest girth and total length were most highly correlated with body mass for ABIR and SRS, respectively (Table 1). As expected, age class had a significant impact (*p* < 0.05) on SMI scores at both ABIR and SRS, and sex had a significant effect at SRS. TTSuV1 infection status had no significant association with SMI at either site.

### 3.5. Genetic Divergence of TTSuV1 over Space

A total of 28 and 24 TTSuV1 sequences obtained from ABIR and SRS, respectively, met our previously described criteria and were included in the genetic distance matrices. Pairwise, geographic distances of the capture locations ranged from 504 to 7479 m and 715 to 19,789 m at ABIR and SRS, respectively, not including pigs caught in the same trap, which had a geographic distance of 0.

The Mantel correlograms for both ABIR and SRS showed a positive Mantel correlation within the first distance class, suggesting the viral genetic sequences were more similar at shorter geographic distances within this distance class (Figure 3). As the first distance class was shorter than the minimum distance between traps at both sites, it exhibited a clear pattern that the viral sequences within traps were significantly similar, but beyond that, the genetic distances appeared to be uncorrelated with geographic distances. At intermediate distances (i.e., 3000–6000 m), the genetic distance of viral sequences from SRS was significantly negatively correlated with geographic distance. That is, as geographic distance increased within this range, the viral genetic distance decreased (more similar), which might be due to the dispersal of maturing males from their family groups.

### 3.6. Phylogenetic Analysis

We first tested substitution patterns in all 52 unique TTSuV1 sequences by plotting the proportions of transitions and transversions in each pair of sequences over genetic distance (Figure 4). Both relationships increased with genetic distance between pairs of sequences, suggesting that these sequences were not saturated, and the phylogenetic signal was not influenced by homoplasy. Xia’s test also indicated there was no substitution saturation for this viral DNA segment as the index of substitution saturation (*I_ss_*) was smaller than the critical value (*I_ss_* < *I_ss.c_*; *p* < 0.01).

The phylogenetic tree in Figure 5 showed separate clades reflecting 4 previously described subtypes of TTSuV1 (TTSuV1a-d) [27]. Together, subtypes 1c and 1d accounted for 80.8% of the 52 sequences typed. Subtype 1d was the most common subtype at ABIR (*n* = 12), while subtype 1c was the most common one at SRS (*n* = 12). There was only one subtype 1a sequence, which was obtained from ABIR. Within subtypes, we found clustering of sequences by site, though there were a few mixtures of sequences from both sites in some clades.

### 3.7. Case Study: Host Relatedness and Virus Transmission at ABIR

#### 3.7.1. Correlation of Genetic Distance Matrices of Hosts and Viruses

We created two alternative matrices from 96 pig samples collected from ABIR to describe genetic differences among wild pigs: a host relatedness matrix and a genetic distance matrix (Appendix A). Of these two approaches, genetic distance, as opposed to relatedness, more clearly demonstrated associations with geographic distance. The Mantel correlogram of host genetic distance compared to the geographic distance indicated that wild pigs in the first distance class (500 m) and at distance classes >1500 m followed a pattern of isolation by distance, i.e., as geographic distance increased, so did pig genetic distance (Figure 6A).

The correlation between pairwise genetic distances of infected hosts and viral genetic distance of TTSuV1 found in those hosts were significantly positively correlated, and the magnitude of correlation decreased with increasing genetic distance between hosts (Figure 6B), i.e., TTSuV1 genetic sequences from wild pigs with smaller genetic distances tended to be more genetically similar. This positive correlation was significant in the first two distance classes.

#### 3.7.2. Inference of Host-To-Host Viral Transmission

The dated phylogeny of TTSuV1 estimated by the Optimized Relaxed Clock model (Figure 7A) had the greatest marginal likelihood compared with the other two models (Appendix A) and was used for inferring the transmission tree (Figure 7B). We identified 10 transmission events where both infector and infectee were pigs sampled at ABIR (Table 3). Six of these inferred virus transmission events occurred within the family group as inferred by capture distance (individuals were captured in the same trap site), the infector-infectee genetic relatedness was ≥0.15, and the average genetic relatedness of these pig pairs was 0.38 ± 0.16. Two transmission events between hog296-hog295 and hog254-hog266 suggested transmission during mating assumed from pairs of pigs having opposite sex, distant genetic relationships, and capture locations that were far apart (>900 m).

## 4. Discussion

We examined the apparent prevalence of TTSuV1 infections among wild pigs in two study sites. We characterized several key features of this virus, including persistence of infection, correlation with the health status of wild pigs, and genetic divergence on the landscape to assess the potential utility of TTSuV1 as a surrogate for future ecological and molecular epidemiological studies of TADs. We also conducted a molecular epidemiological case study to determine how analysis of this virus could inform transmission studies. The results revealed a distinct pattern of direct transmission from pig to pig occurring within and between family groups.

### 4.1. Evaluation of TTSuV1 as a Surrogate TAD

TTSuV1 was prevalent in wild pigs at both study sites. Comparison of infection rates between age classes indicated that adult pigs had a higher prevalence and were more likely to be infected with TTSuV1 than juveniles at one site but not the other. Other studies of directly transmitted pathogens have described increasing infection rates with the age of domestic pigs [64,65], but this outcome depends on herd management and demographic stratification.

Although we were able to recapture 14 individuals and found many of them to be infected at both capture time periods, in most cases, the viral sequences were too dissimilar to be consistent with a persistent infection. The observed sequence variations in recaptured pigs may be influenced by the potential reactivation of the virus, a phenomenon known to occur within the *Anelloviridae* family under certain immunologic conditions [66]. This possibility should be considered when interpreting differences in sequences obtained from the same individuals over time. The limited recapture data did not allow us to address the question of how long TTSuV1 persists in the host, and further studies are needed.

To date, TTSuV1 has been considered non-pathogenic and not associated with any diseases or symptoms [24]. Previous studies reported no difference in rates of stillbirth, mortality rates in piglets, or clinical signs of TTSuV1 [65,67]. In this study, we found that TTSuV1 infection did not impact the host body condition, further supporting the above conclusion. The low impact of this virus on the fitness of wild pigs enables researchers to safely handle the virus in situ and in laboratory settings as a surrogate to study the potential spatial spreading dynamics of TADs like African swine fever virus (ASFV), many of which are highly pathogenic and regulated with stringent biosafety requirements.

TTSuV1 demonstrated high genetic diversity within the relatively small landscapes of our study sites, especially in ABIR, where all four subtypes of TTSuV1 were identified. Subtypes 1c and 1d were dominant in the sequenced samples used for constructing the phylogenetic tree. Four subtype 1d sequences (3 from ABIR and 1 from SRS) were unique within the subtype 1d clade and clustered in a sub-clade in the tree. Future sampling and more complete sequencing of these variants are warranted.

It is common to find co-infections with multiple TTSuV species within pigs [31,33]; however, to our knowledge, only one other study has described the presence of multiple variants of TTSuV1 present in a pig simultaneously (TTSuV1-TTSuV1 co-infection) [68]. In this study, we report the presence of multiple TTSuV1 variants within the same tissue of a host for the first time. Multiple variants within a host could arise from either multiple transmission events to create a true co-infection or via the evolution of quasi-species within a host where genetically related but distinct variants or mutants of a virus are generated within a single infected host that creates a cloud of closely related viral genomes termed the quasi-species [69].

### 4.2. A Molecular Epidemiological Case Study

The virus was detected in oral, nasal, and genital samples, and it was most consistently found in whole blood. These findings suggest that transmission of the virus could occur via multiple routes, including nose-to-nose, fecal-oral, and sexual transmission. The presence of TTSuV1 DNA in all types of samples collected in this study supports previous findings on the importance of horizontal transmission for the virus. Our viral transmission data suggested that direct transmission frequently occurred between genetically related pigs from the same family social group. In addition, transmission events between sexually mature pigs that had distant genetic relationships and were captured at geographically distant locations suggest the potential for additional transmission routes for TTSuV1, including sexual or fecal-oral transmission. In swine, transmission of TTSuVs occurs through both horizontal and vertical routes, with fecal-oral transmission considered to be the most important in viral dissemination [70]. The DNA of TTSuV has been found in nasal and fecal samples from week-old pigs, and prevalence increases with the age of the hosts [65]. Vertical transmission has also been documented in the presence of TTSuV1 DNA in the blood and tissue of fetuses and in the semen and colostrum of adult pigs [70,71,72]. All these transmission routes overlap with that of ASFV and classical swine fever virus (CSFV), with horizontal transmission as the primary mode of transmission [73,74].

Wild pigs are known for their social behaviors and often form matrilineal social groups, known as sounders, to forage and provide protection from predators [75,76]. A sounder typically consists of one or more adult females (sows) and their offspring, which can include piglets of varying ages [34]. The size of a sounder can vary based on factors such as habitat, forage availability, time of year, and population density [77]. The close proximity of individuals within a given sounder could lead to increased direct contact among members of the social group [78] and, consequently, a higher probability of spreading pathogens like TTSuV1. We hypothesized that TTSuV1 would be transmitted within sounders more frequently than among sounders, which was borne out by the results that the viruses were more genetically similar between closely related animals. Theoretical models have predicted this relationship between host and virus in pigs [78,79], and our study empirically demonstrates those relationships.

The Mantel correlograms corroborate the molecular epidemiological signal of direct transmission among relatives. At shorter distances, pigs and their viruses were genetically similar, yet at intermediate distances, the relationship was reversed. Viral genetic distance had a significantly negative correlation with geographic distance (i.e., the farther the geographic distance, the more genetically similar the viral sequences). One explanation for this signal at intermediate distances is due to the dispersal of mature male wild pigs from sounders. Male wild pigs are generally solitary animals and roam their territories independently. They only have a temporary association with sounders during growth to maturity and in the breeding season, and in both circumstances, they do not remain within a sounder. If they were infected with TTSuV1 during their stay in sounders and then dispersed, the geographic distances would be large, but the viral genetic distance would be small. Further analyses with animal movement data are needed to test this hypothesis.

Given the above findings, we suggest that TTSuV1 can be used as a surrogate TAD, which will contribute to the empirical validation of theoretical models of TAD transmission in wild pigs. For example, Yang et al. [78] constructed a contact network using proximity logger data and GPS data of wild pigs at ABIR and SRS and simulated the effects of different management practices on the contact network as well as the risk of ASF establishment among wild pigs. They found that low-intensity removal (<5.9% of the population) did not alter the contact structure of wild pigs but reduced the predicted *R_0_* of the ASF virus by ~42–78%. However, concentrated livestock feeding or baiting of wild pigs, a method used to attract wild pigs or game species for recreational hunting, raised the establishment risk of ASF virus by approximately 27–33% as a result of increased indirect contact rates at feed sites. These models create hypotheses for the transmission dynamics of TADs among wild pigs; molecular epidemiological analysis of TTSuV1 would provide empirical data to test those hypotheses. In addition, it could also be used to assess how much transmission occurs among wild pigs and domestic populations (commercial or backyard), which is one of the major concerns in TAD prevention and control.

Surrogate TADs could also be used to empirically test the efficacy of population management actions on disease dynamics. For example, reducing the host population density by culling/hunting has been one of the common tools for controlling disease transmission among wildlife populations [80]; however, the effects of culling on limiting the spread of disease are ambiguous. Though often a successful management tool for wildlife diseases [81,82,83], culling a wildlife population has the potential to trigger unforeseen alterations in host movement patterns or territoriality, which could increase the risk of transmission through increased contact or further movement [78,80,82]. Culling European badgers (*Meles meles*) was part of a strategy to control bovine tuberculosis (TB) in Great Britain. This management strategy not only failed to control the disease but may have increased TB incidence in cattle populations [82]. Empirically testing these management actions with surrogate TADs may provide additional insights into the multifaceted dynamics of pathogen transmission.

There were several limitations in this study. Though the phylogenetic tree reconstructed from TTSuV1 isolates from both sites demonstrated a clear separation of four TTSuV1 subtypes, uncertainty was observed through low support values in the tree. To further characterize its genomic features and determine the applicability of TTSuV1 in molecular epidemiological models, whole genome sequences may be necessary in future studies. The transmission tree inferred in our study provided a useful way to understand the transmission dynamics of the virus among wild pigs with varying relatedness. Due to the unavailability of TTSuV generation time data, we used that of PCV2; thus, the results should be interpreted with a caveat that distinctions between the average generation times of TTSuV1 and PCV2 likely exist, though they are both small porcine circular DNA viruses. It is worthwhile to design experiments to estimate the generation time of TTSuV1 to better facilitate the utility of it as a surrogate in future modeling work.

In conclusion, TTSuV1 was prevalent and diverse in wild pigs caught at both study sites. At ABIR, where we conducted the case study, hosts demonstrated nonrandom clustering of related individuals in family groups at distances <1000 m, which influenced the pattern of transmission across the landscape. From the transmission tree, we inferred direct viral transmission within and between family groups, and there is evidence that other routes of transmission may be occurring based on our observations of viral shedding of TTSuV1 in nasal, buccal, and genital mucous. Our findings indicate that molecular epidemiological analysis of TTSuV1 provides insights into the relationship between animal behavior and transmission dynamics in a free-ranging wild pig population in the southeastern USA. Moreover, TTSuV1 has the necessary characteristics to make it a useful surrogate pathogen for studying the transmission of pathogens in wild pigs, with the potential to inform and improve disease control among wildlife populations.

## Figures and Tables

**Figure 1 viruses-16-01397-f001:**
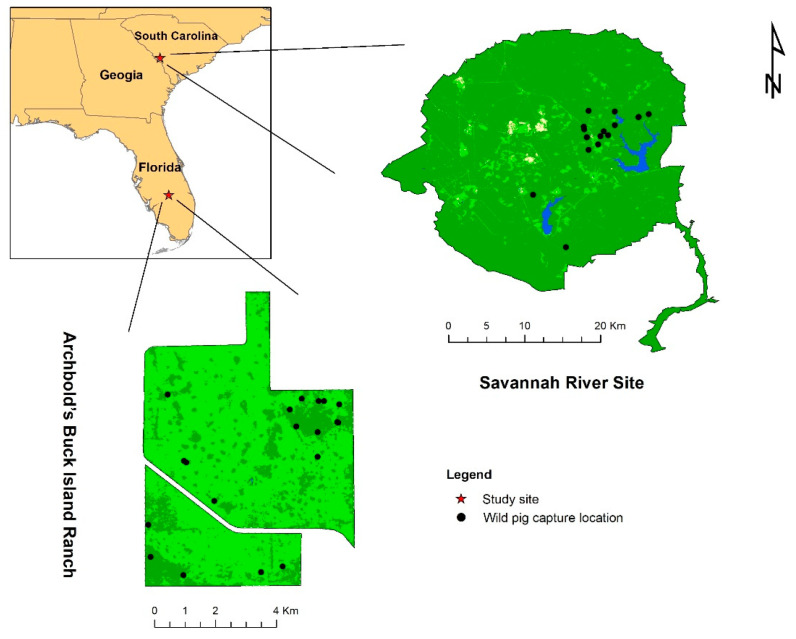
Study sites and live capture locations of wild pigs were included in this study.

**Figure 2 viruses-16-01397-f002:**
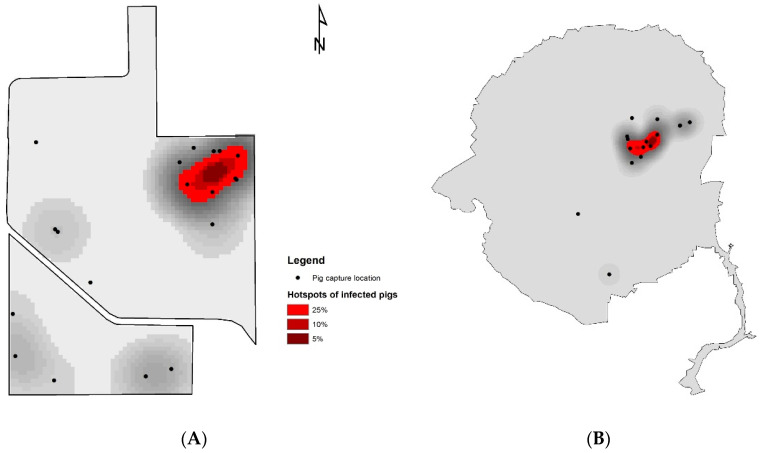
Kernel density estimation (KDE) of wild pigs infected with TTSuV1 at Archbold’s Buck Island Ranch (**A**) and Savannah River Site (**B**). Black dots represent the capture locations of wild pigs. The sample size at each capture location ranges from 1 to 22 at ABIR and from 1 to 18 at SRS. The red color ramp represents the hotspots of infected wild pigs based on 5%, 10%, and 25% of KDE values.

**Figure 3 viruses-16-01397-f003:**
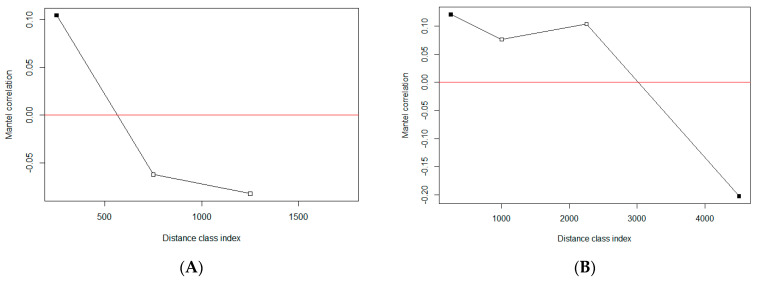
Mantel correlation analysis of the genetic and geographic distances of TTSuV1s obtained from two study sites. (**A**) Archbold’s Buck Island Ranch; (**B**) Savannah River Site. The *x*-axis represents the geographic distances divided into several distance classes, and the *y*-axis represents the mantel correlation values for each distance class. A solid square represents a statistically significant (*p* < 0.05) Mantel correlation.

**Figure 4 viruses-16-01397-f004:**
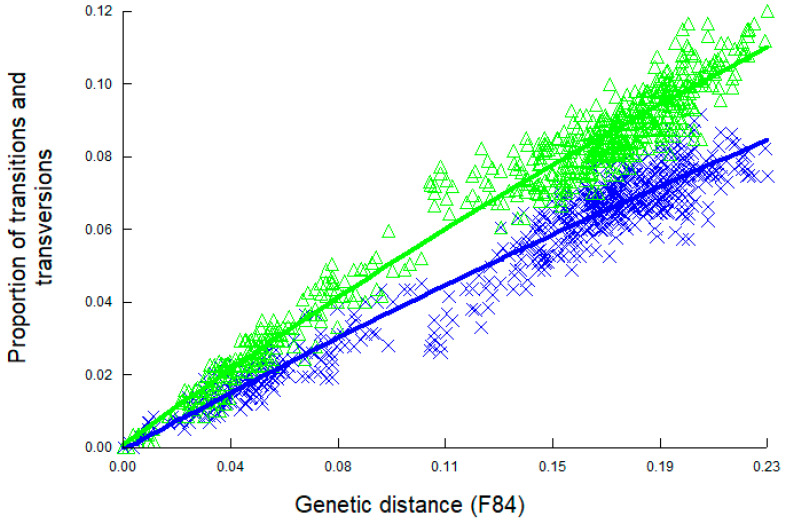
Substitution saturation plot for TTSuV1 sequences obtained in this study. The plot displays the observed proportion of transitions (blue crosses) and transversions (green triangles) between each pair of sequences (*n* = 52) over genetic distance based on the F84 substitution model. Lines were generated using the best fit for observed data.

**Figure 5 viruses-16-01397-f005:**
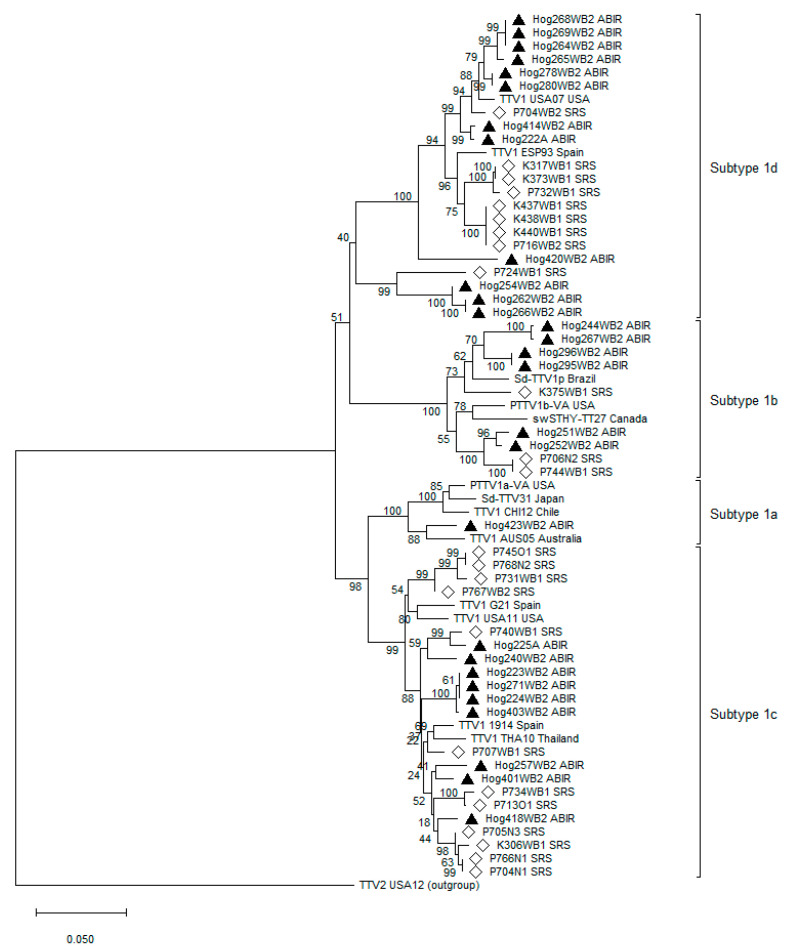
A phylogenetic tree was constructed from 52 TTSuV1 sequences obtained in this study. Solid triangles and hollow diamonds represent sequences from Archbold’s Buck Island Ranch and Savannah River Site, respectively. Sequences without a triangle or diamond are reference sequences. The tree was rooted using a TTSuV2 sequence as the outgroup.

**Figure 6 viruses-16-01397-f006:**
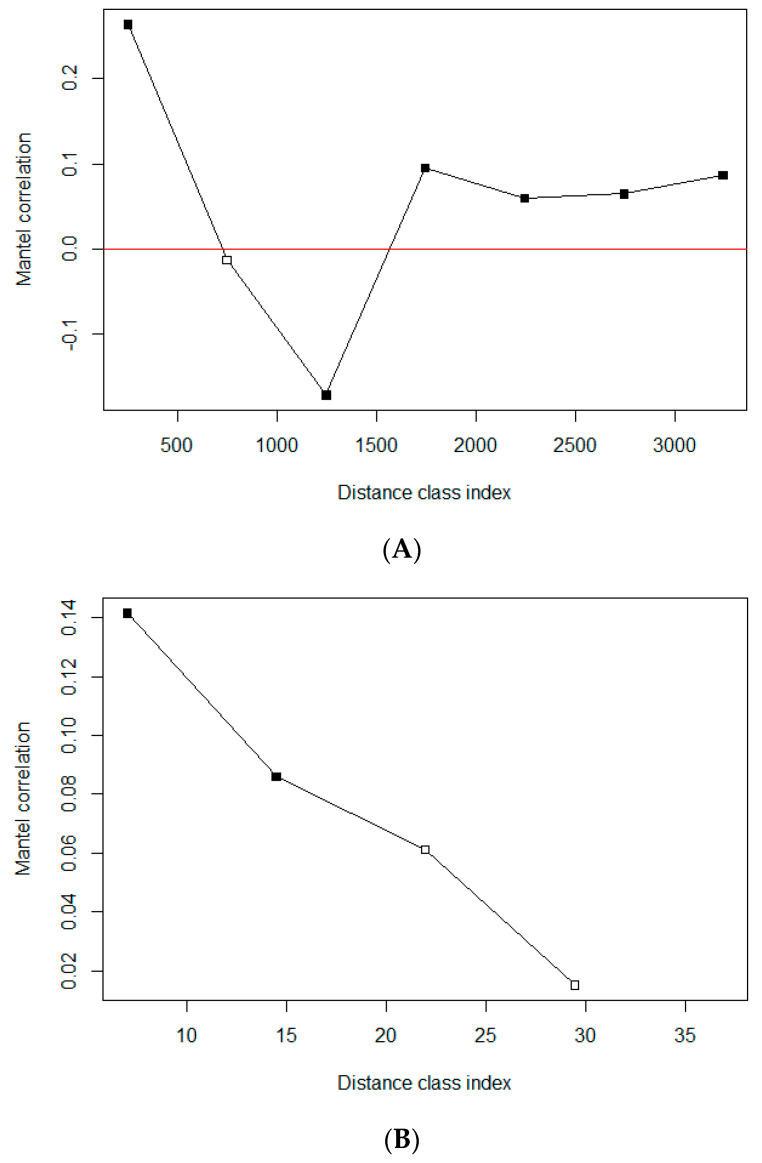
Mantel correlogram analysis of (**A**) host genetic distance vs. geographic distance (*x*-axis); (**B**) viral genetic distance vs. host genetic distance (*x*-axis) at ABIR. The *y*-axis represents the Mantel correlation values for each corresponding distance class at the *x*-axis. A solid square represents a statistically significant (*p* < 0.05) Mantel correlation.

**Figure 7 viruses-16-01397-f007:**
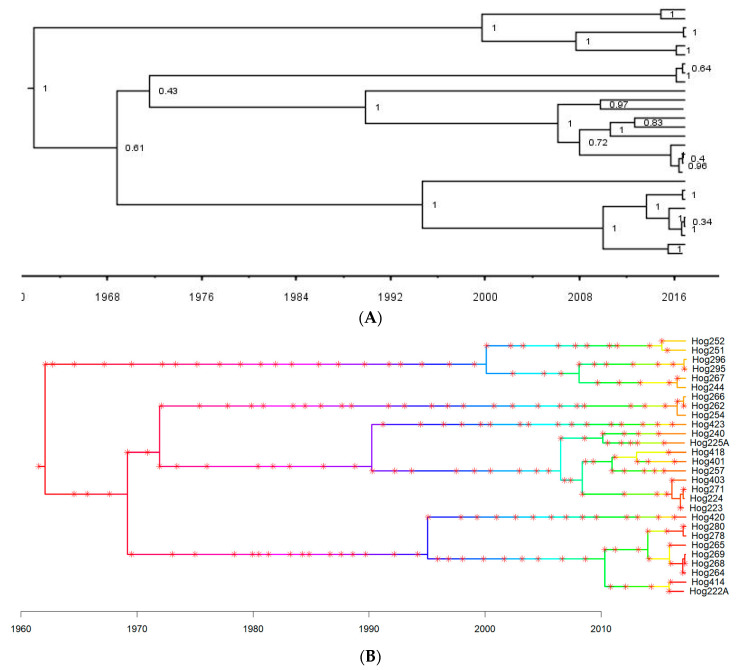
The dated phylogeny and transmission tree of TTSuV1 isolated at Archbold’s Buck Island Ranch, 2017–2018. (**A**). Maximum clade credibility (MCC) tree estimated from 28 TTSuV1 sequences isolated from wild pigs at ABIR. (**B**). The transmission tree is inferred from the dated phylogeny of TTSuV1 at ABIR. Varying colors of branches represent different individuals, both sampled and unsampled, in the transmission chains. The tips are the sampled individuals in this study, and unsampled individuals are those inferred by the algorithm based on the phylogeny of viruses and other transmission-related parameters. They act as missing transmission links in the transmission tree. Red asterisks represent inferred transmission events.

**Table 1 viruses-16-01397-t001:** Description of wild pig body measurements and their correlation to body mass.

Site	Body Measurement	Description	Correlation to Body Mass ^$^
ABIR *	Neck circumference	Circumference at neck	0.926
	Chest Girth	Girth just behind forelegs	0.968
	Wither length	Girth just in front of back legs	0.934
	Body length	Base of tail to base of ears	0.900
	Total length	Base of tail to tip of nose	0.898
SRS ^#^	Chest Girth	Girth behind forelegs	0.936
	Total length	Base of tail to tip of nose	0.939

* Archbold’s Buck Island Ranch. ^#^ Savannah River Site. ^$^ Pearson’s correlation coefficient.

**Table 2 viruses-16-01397-t002:** Prevalence of TTSuV1 in wild pigs across sample types and field sites.

Site		Sample Type				
		Blood	Oral	Nasal	Genital	Total
	No.	97	102	102	100	102
ABIR	Positive #*	40	0	0	0	40
	Prevalence ^$^	41.2% _a_ (31.9–51.1%)	0.0% _a_ (0–3.6%)	0.0% _a_ (0–3.6%)	0.0% _a_ (0–3.7%)	39.2% _a_ (30.3–48.9%)
	No.	87	88	88	88	88
SRS	Positive #	32	6	6	1	34
	Prevalence	36.8% _a_ (27.4–47.3%)	6.8% _b_ (3.2–14.1%)	6.8% _b_ (3.2–14.1%)	1.1% _a_ (1.9–6.1%)	38.6% _a_ (29.1–49.0%)

* Number of positive cases. ^$^ Prevalence with different subscripts differs significantly (*p* < 0.05) between sites. All confidence intervals in brackets were calculated using Wilson confidence limits at the 95% confidence level.

**Table 3 viruses-16-01397-t003:** Transmission events inferred by the transmission tree of TTSuV1-infected wild pigs at Archbold’s Buck Island Ranch.

Sample ID	Sex	Age	Infectee	Sex	Age	Distance (m)	Relatedness
hog296	Male	2Y	hog295	Female	2Y	911.3	0.0006
hog244	Female	3Y	hog267	Female	1Y	653.6	0.08
hog266	Female	1Y	hog262	Male	1Y	0	0.23
hog254	Male	1Y	hog266	Female	1Y	1450.3	0.01
hog224X	Female	1Y	hog271	Female	1Y	0	0.48
hog224X	Female	1Y	hog223X	Female	2Y	0	0.53
hog224X	Female	1Y	hog403	Female	1Y	660.1	0.09
hog278	Female	2Y	hog280	Male	3M	0	0.15
hog269	Female	1Y	hog268	Female	1Y	0	0.45
hog269	Female	1Y	hog264	Male	1Y	0	0.45

## Data Availability

The metadata of wild pig samples that support the findings of this study are available from the corresponding author upon reasonable request. All viral sequence data with metadata are available in NCBI GenBank.

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
