# Peer review of "Torque Teno Sus Virus 1: A Potential Surrogate Pathogen to Study Pig-Transmitted Transboundary Animal Diseases"

_viruses, 2024, doi:10.3390/v16091397_

Round 1

Reviewer 1 Report

Comments and Suggestions for Authors

Xiaolong Li and colleagues explored a surrogate pathogen system for in situ studies of pathogen dynamics could to shed light on the epidemiology and transmission dynamics of transboundary animal diseases (TADs) among wild pigs (Sus scrofa) will aid in preventing the introduction or containment of TADs among wild populations. Thus, the authors assessed the suitability of Torque teno sus virus 1 (TTSuV1) to serve as a surrogate pathogen in pigs in the Areas of Archbold’s Buck Island Ranch in Florida and Savannah River Site in South Carolina. Prevalence was high at both study areas (40%) and phylo-genetic analyses revealed high levels of genetic variability within and between study sites. Then the authors investigated pairwise host relatedness and geographic distance and found highly correlated to pairwise viral genetic similarity. Molecular epidemiological analyses revealed a distinct pattern of direct transmission from pig to pig occurring within and between family groups.

Their hypotheses posited that closely related pigs, especially females and juveniles, would exhibit spatial proximity given their matriarchal social structure and that this spatial proximity would lead to transmission within family groups.

The study is of interest but there are some point that have to be clear explained to inprove the paper.

Main points

1-       Abstract and text: the authors reported the word “in situ”. According my point of view the term is confusing. What is the meaning of? It could be better change or delete it from the text. Otherwise, the authors should explain it.

2-       Page 2, lines 17-20: the authors reported “…pathogen system for in situ studies of pathogen dynamics would be ideal to better predict how these pathogens may circulate and be maintained among wild populations, how TADs may spillover into commercial operations, and how disease spread may be mitigated”. How this assertion could be completely explained by the use of TTSuV1 in all part? The TTSuV1 transmission has the same modalities of all the TADs related pathogens? Moreover, the TTSuV1 variability is high. Have the TADs related pathogens similar variability? All these characteristics could be relevant in the investigation of a model of virus transmission surrogate. The authors should explain it better or explain potential limitation in the conclusion.

3-       Page 2, line 40: the authors have to modify the number of the Anelloviridae genera according the recent “Taxonomic update for mammalian anelloviruses (family Anelloviridae)” study.

4-       Page 3, materials and methods: The authors stated “Biological samples were stored at -20 °C until ex-tracted”. How many time after the collection they performed the molecular investigation (Viral DNA was extracted, quantification, PCR)? The conservation at -20°C at long time could be deleterious for the samples.

5-       Page 4, Sequence analyses: “All PCR products of the appropriate size were submitted….” The authors should report the sequence used (position and gene of interest).

6-       Page 5 (2.3.2. Persistence of TTSuV1 infection): what is the meaning of “…then they were considered the same variant and therefore a serial infection”. The authors should change this assertion with “persistent infection” instead of “serial infection”.

7-       What is the meaning of the difference obtained in TTSuV1 in the two areas (Areas of Archbold’s Buck Island Ranch in Florida and Savannah River Site in South Carolina) for the difference in blood positivity and age related prevalences? The TTSuV1 could have different transmission in the two areas? Moreover, how these differences could influence the model?

8-       Page 16 (discussion, line 17-20): In the estimation of recapture pigs with the assumption of the inconsistence of  a persistent infection for the dissimilar sequence obtained, it could be considered the potential reactivation of the virus infection for particular immunologic condition. Iin this context it is well known the Anelloviridae reactivation as well as re-infection.

Minor points

1-       Page 2, line 39: add reference 18.

2-       Page 4, line 9: the sequence of primer used has to be added in the text. Moreover, why it was used a one-step PCR?

3-       Table 3: in the table the first column should has a name.

4-       All typos should be corrected.

Author Response

Reviewer 1

Xiaolong Li and colleagues explored a surrogate pathogen system for in situ studies of pathogen dynamics could shed light on the epidemiology and transmission dynamics of transboundary animal diseases (TADs) among wild pigs (Sus scrofa) will aid in preventing the introduction or containment of TADs among wild populations. Thus, the authors assessed the suitability of Torque teno sus virus 1 (TTSuV1) to serve as a surrogate pathogen in pigs in the Areas of Archbold’s Buck Island Ranch in Florida and Savannah River Site in South Carolina. Prevalence was high at both study areas (40%) and phylo-genetic analyses revealed high levels of genetic variability within and between study sites. Then the authors investigated pairwise host relatedness and geographic distance and found highly correlated to pairwise viral genetic similarity. Molecular epidemiological analyses revealed a distinct pattern of direct transmission from pig to pig occurring within and between family groups.

Their hypotheses posited that closely related pigs, especially females and juveniles, would exhibit spatial proximity given their matriarchal social structure and that this spatial proximity would lead to transmission within family groups.

The study is of interest but there are some points that have to be clear explained to improve the paper.

Response: We appreciate the reviewer’s interest in our study and have addressed the specific points raised to enhance the clarity and overall quality of the paper.

Main points

1 Abstract and text: the authors reported the word “in situ”. According to my point of view the term is confusing. What is the meaning of? It could be better change or delete it from the text. Otherwise, the authors should explain it.

Response: Thank you for your suggestion. We have replaced the term "in situ" with "in wild free-ranging populations" throughout the paper to enhance clarity and avoid any potential confusion.

2 Page 2, lines 17-20: the authors reported “…pathogen system for in situ studies of pathogen dynamics would be ideal to better predict how these pathogens may circulate and be maintained among wild populations, how TADs may spillover into commercial operations, and how disease spread may be mitigated”. How this assertion could be completely explained by the use of TTSuV1 in all part? The TTSuV1 transmission has the same modalities of all the TADs related pathogens? Moreover, the TTSuV1 variability is high. Have the TADs related pathogens similar variability? All these characteristics could be relevant in the investigation of a model of virus transmission surrogate. The authors should explain it better or explain potential limitation in the conclusion.

Response: Thank you for your comment. In the Introduction, we emphasized the broader need for a surrogate pathogen system that could potentially address issues like circulation of TAD-related pathogens in wild populations, spillover into commercial operations, and disease mitigation. However, we did not intend to suggest that the TTSuV1 system studied in this paper would definitively answer these specific questions. Instead, our study explores the potential of TTSuV1 as a surrogate and discusses its possible applications in this field, as outlined in the Discussion section.

We have now modified the abstract and introduction to be more circumspect about how a surrogate system could be used.

3 Page 2, line 40: the authors have to modify the number of the Anelloviridae genera according to the recent “Taxonomic update for mammalian anelloviruses (family Anelloviridae)” study.

Response: We have updated the number of Anelloviridae genera in our revised manuscript to reflect the most recent taxonomy as per the study you referred to and cited that paper in the manuscript to ensure the information is accurately referenced.

4 Page 3, materials and methods: The authors stated “Biological samples were stored at -20 °C until ex-tracted”. How many time after the collection they performed the molecular investigation (Viral DNA was extracted, quantification, PCR)? The conservation at -20°C at long time could be deleterious for the samples.

Response: DNA extraction occurred 24 – 48 months after the biological samples were stored at -20 °C. We have added that description in the Methods. As there were no freeze-thaws until extraction and our samples amplified well, we do not think degradation was a concern

5 Page 4, Sequence analyses: “All PCR products of the appropriate size were submitted….” The authors should report the sequence used (position and gene of interest).

Response: Thanks for your suggestion. The related information was mentioned in the previous paragraph: “A one-step PCR assay was conducted using primers previously described (Cortey et al., 2012) to detect TTSuV1 based on a 678 base pair (bp) sequence that encompasses portions of the untranslated region (UTR) and open reading frame (ORF) 1 and the entire ORF2.”

6 Page 5 (2.3.2. Persistence of TTSuV1 infection): what is the meaning of “…then they were considered the same variant and therefore a serial infection”. The authors should change this assertion with “persistent infection” instead of “serial infection”.

Response: We have made the change accordingly in the revised manuscript as the reviewer suggested.

7 What is the meaning of the difference obtained in TTSuV1 in the two areas (Areas of Archbold’s Buck Island Ranch in Florida and Savannah River Site in South Carolina) for the difference in blood positivity and age related prevalences? The TTSuV1 could have different transmission in the two areas? Moreover, how these differences could influence the model?

Response: These differences in blood positivity and age-related prevalence could reflect distinct ecological conditions, host population dynamics, or management strategies unique to each site, which warrants further studies to explore these factors and understand how they influence the observed variations. However, our goal in comparing the two sites in this study was to investigate whether there were any differences in virus prevalence and diversity between two wild pig populations. Our results showed no significant difference between the two sites, indicating that TTSuV1 is ubiquitous across different environments..

We only used data from one site as a molecular epidemiological case study, so the differences between the two sites will not influence the model. However, future studies are needed to evaluate how such differences could affect the model's applicability across different environments. We clarify this point in the concluding paragraph of the study.

8 Page 16 (discussion, line 17-20): In the estimation of recapture pigs with the assumption of the inconsistence of a persistent infection for the dissimilar sequence obtained, it could be considered the potential reactivation of the virus infection for particular immunologic condition. Iin this context it is well known the Anelloviridae reactivation as well as re-infection.

Response: We acknowledge that the Anelloviridae family is known for viral reactivation under certain immunologic conditions. In our estimation of recaptured pigs with dissimilar sequences, it is possible that the differences observed could be attributed to the reactivation of the virus rather than new infections. We have included the following in the Discussion:

“The observed sequence variations in recaptured pigs may be influenced by the potential reactivation of the virus, a phenomenon known to occur within the Anelloviridae family under certain immunologic conditions (Sabbaghian et al., 2024). This possibility should be considered when interpreting differences in sequences obtained from the same individuals over time.”

Minor points

1 Page 2, line 39: add reference 18.

Response: We have added the reference Cortey et al. 2011 to that specific position.

2 Page 4, line 9: the sequence of primer used has to be added in the text. Moreover, why it was used a one-step PCR?

Response: The sequences of the primers used have been added to the text for clarity. We opted for a one-step PCR to streamline the amplification process, reducing the potential for contamination and saving time by combining reverse transcription and amplification in a single reaction. This approach also increases the efficiency and consistency of the results.

3 Table 3: in the table the first column should has a name.

Response: We have added a column name to the first column of Table 3.

4 All typos should be corrected.

Response: We have thoroughly reviewed the manuscript and corrected all typos to ensure clarity and accuracy.

Reviewer 2 Report

Comments and Suggestions for Authors

The manuscript by Li et al shows a study on TTSuV1 among wild pigs at two regions in the USA. In search for a marker that visualizes the chance for transboundary animal diseases for pigs, the authors investigated TTSuV1 as a surrogate pathogen. The study has an excellent design with highly relevant testing and appropriate analysis of the TTSuV1 PCR product sequencing results. The study is also clearly written, and the main conclusion that TTSuV1 sequencing is suitable for molecular epidemiological analyses to monitor transmission dynamics in wild pigs, is a valid conclusion. I only have one suggestion for improvement, see comment see below.

 Comment:

Paragraph 2.3.2. In this paragraph of the Materials and Methods the authors write the following: “If viral sequences from a recaptured host had ≥95% PI, then they were considered the same variant and therefore a serial infection” The authors may also consider that the repeatedly found variant was the result of persistent infection in the pig. Anellovirus variant infections in human adults are known to be chronic (monitored >30 years) with variants seldomly cleared with time. There is no reason to assume that this would not also occur in pigs.

Author Response

Reviewer 2

The manuscript by Li et al shows a study on TTSuV1 among wild pigs at two regions in the USA. In search for a marker that visualizes the chance for transboundary animal diseases for pigs, the authors investigated TTSuV1 as a surrogate pathogen. The study has an excellent design with highly relevant testing and appropriate analysis of the TTSuV1 PCR product sequencing results. The study is also clearly written, and the main conclusion that TTSuV1 sequencing is suitable for molecular epidemiological analyses to monitor transmission dynamics in wild pigs, is a valid conclusion. I only have one suggestion for improvement, see comment see below.

Response: We appreciate your positive assessment of our study.

 Comment:

Paragraph 2.3.2. In this paragraph of the Materials and Methods the authors write the following: “If viral sequences from a recaptured host had ≥95% PI, then they were considered the same variant and therefore a serial infection” The authors may also consider that the repeatedly found variant was the result of persistent infection in the pig. Anellovirus variant infections in human adults are known to be chronic (monitored >30 years) with variants seldomly cleared with time. There is no reason to assume that this would not also occur in pigs.

Response: Thank you for this insightful comment. We agree that the repeatedly found variant may indeed be the result of a persistent infection in the pig. We have replaced “serial infection” with “persistent infection” in that sentence.

Round 2

Reviewer 1 Report

Comments and Suggestions for Authors

The authors have substantially improuved the manuscript that are now suitable for publication with the only change the format of reference citing (in the text reference should be cited by number placed in square brackets as they were in the first version). 

Author Response

Dear Editor:

As requested by the reviewer, we have now changed the citations to a numerical format.

Sincerely,

Samantha Wisely